# Has the Removing of the Mesentery during Ileo-Colic Resection an Impact on Post-Operative Complications and Recurrence in Crohn’s Disease? Results from the Resection of the Mesentery Study (Remedy)

**DOI:** 10.3390/jcm11071961

**Published:** 2022-04-01

**Authors:** Michela Mineccia, Giovanni Maconi, Marco Daperno, Maria Cigognini, Valeria Cherubini, Francesco Colombo, Serena Perotti, Caterina Baldi, Paolo Massucco, Sandro Ardizzone, Alessandro Ferrero, Gianluca M. Sampietro

**Affiliations:** 1Division of General and Oncologic Surgery, Ospedale Mauriziano Umberto I, Largo Turati 62, 10128 Torino, Italy; mmineccia@mauriziano.it (M.M.); valeria.cherubini@unito.it (V.C.); sperotti@mauriziano.it (S.P.); pmassucco@mauriziano.it (P.M.); aferrero@mauriziano.it (A.F.); 2Division of Gastroenterology, ASST Fatebenefratelli-Sacco, via G.B. Grassi 74, 20157 Milano, Italy; giovanni.maconi@unimi.it (G.M.); sandro.ardizzone@asst-fbf-sacco.it (S.A.); 3Division of Gastroenterology, Ospedale Mauriziano Umberto I, Largo Turati 62, 10128 Torino, Italy; mdaperno@mauriziano.it; 4Division of General Surgery, ASST Fatebenefratelli-Sacco, Ospedale, via G.B. Grassi 74, 20157 Milano, Italy; maria.cigognini@unimi.it (M.C.); francesco.colombo@asst-fbf-sacco.it (F.C.); caterina.baldi@unimi.it (C.B.); 5Division of General and HPB Surgery, ASST Rhodense, Rho Memorial Hospital, Corso Europa 250, 20017-Rho Milano, Italy

**Keywords:** IBD, Crohn’s disease, surgery, resection, anastomosis, mesentery, complications, surgical recurrence

## Abstract

Some evidence suggests a reduction in clinical and surgical recurrence after mesenteric resection in Crohn’s Disease (CD). The aim of the REsection of the MEsentery StuDY (Remedy) was to assess whether mesenteric removal during surgery for ileocolic CD has an impact in terms of postoperative complications, endoscopic and ultrasonographic recurrences, and long-term surgical recurrence. Among the 326 patients undergoing primary resection between 2009 and 2019 in two referral centers, in 204 (62%) the mesentery was resected (Group A) and in 122 (38%) it was retained (Group B). Median follow-up was 4.7 ± 3 years. Groups were similar in the peri-operative course. Endoscopic and ultrasonographic recurrences were 44.6% and 40.4% in Group A, and 46.7% and 41.2% in Group B, respectively, without statistically significant differences. The five-year time-to-event estimates, compared with the Log-rank test, were 3% and 4% for normal or thickened mesentery (*p* = 0.6), 2.8% and 4% for resection or sparing of the mesentery (*p* = 0.6), and 1.7% and 5.4% in patients treated with biological or immunosuppressants versus other adjuvant therapy (*p* = 0.02). In Cox’s model, perforating behavior was a risk factor, and biological or immunosuppressant adjuvant therapy protective for surgical recurrence. The resection of the mesentery does not seem to reduce endoscopic and ultrasonographic recurrences, and the five-year recurrence rate.

## 1. Introduction

Mesentery thickening, wrapping fat, and lymph node enlargement have been considered, since their first description, a landmark of the terminal ileitis that became known as Crohn’s disease (CD) [1].

However, the mesentery of bowel segments affected by CD has received fluctuating attention over the years and has been considered by many as a bystander of the pathological events occurring in the mucosal and underlying layers [2,3,4,5]. In recent decades, mesenteric adipose tissue has been proposed as being involved in metabolic, endocrine, and immune disease, secreting cytokines, and hormones, and playing a dysregulatory role in systemic inflammatory syndromes [6,7,8]. Notwithstanding, since during surgery a pathologic mesentery could heavily bleed when divided, the attitude of most surgeons has been to retain it. Some authors recently suggested that the mesentery is not only part of the pathologic events that lead to bowel damage typical of CD, but also has prognostic implications during and after surgical treatment [9,10,11,12,13,14]. On the contrary, a large, retrospective analysis on the prevalence and significance of mesentery thickening and lymph node enlargement did not show any effect on long-term recurrence, and in the 2020 guidelines of the Italian Society of Colorectal Surgery (SICCR), mesentery removal during intestinal resection for CD was not recommended [15,16]. To date, there are no data in the literature that provide reliable information on the need for removing the mesentery to obtain a lower surgical recurrence rate.

The aim of the present study was to assess whether mesenteric removal during primary ileocolic resection for CD plays a role in terms of postoperative complications, endoscopic and ultrasonographic recurrences, and long-term surgical recurrences. 

## 2. Patients and Methods

Mauriziano Hospital (MH), Torino, and ASST Fatebenefratelli-Sacco (FSH), Milano, are tertiary care hospitals for the treatment of Inflammatory Bowel Diseases (IBD) in Italy. The decision to operate was taken in both hospitals during multidisciplinary meeting after complete diagnostic work-up and confirmed diagnosis of CD [15,17,18,19,20]. Patients’ pre-operative optimization was carried out with nutritional support, preoperative abscess percutaneous drainage when necessary, and steroid and biological drugs discontinuation when feasible [15,20,21]. Postoperative complications were stratified using the Clavien-Dindo classification, where serious complications are classified as Grade III and IV, and Grade V corresponds to the patient’s death [22,23]. All of the patients were classified with the Montreal Classification for CD and were inserted in the prospectively maintained surgical databases for IBD patients present in both hospitals that included many variables describing history, pre-operative characteristics, intra-operative and pathological findings, post-operative course, and long-term follow-up [15,19,20,24,25,26,27,28]. There were no classifications in the literature for mesentery thickening of CD patients. As previously reported, the mesentery, a tributary of the diseased bowel segment, was classified as thickened when the thickness was more than 5 mm, with or without fat wrapping, and lymphopathy was defined as the presence of 3 or more lymph nodes, greater than 4 mm. Data were prospectively collected in the databases of both centers based on the large surgical experience on this field, in the absence of specific indications in the literature [15]. 

In the case of ileocolic resection for complicated CD, the attitude toward the mesentery at MH, whether it was normal, thickened, with or without lymph nodes, was dividing it flush with the intestine and leaving it retained, while at FSH was to perform ligation of the vessels not as proximal as in cancer but at D2-level, so that the whole part of the thickened mesentery and lymph nodes were dissected and removed [15]. 

Endoscopic and ultrasonographic post-operative recurrences were evaluated between 6 and 12 months after surgery, and follow-up was scheduled based on the adjuvant therapy undertaken. Endoscopic findings were classified according to the Rutgeerts’ score (from i0 to i4), where remission was defined as a score of i0 or i1 and recurrence with a score from i2 to i4 [29]. As previously reported, ultrasonographic recurrence was defined as a bowel wall thickening ≥4 mm at the site of the previous ileo-colic anastomosis [30,31]. Surgical recurrence was defined as the needing of a new surgical procedure, decided during multidisciplinary meeting, due to complicated CD in the same site of the previous ileocecal resection or any other intestinal segment. All the patients gave their informed consent for the surgical procedure and data auditing.

## 3. The REMEDY Resection of the Mesentry Study

All the consecutive, unselected patients with a single location of CD localized to the terminal ileum, operated on between January 2009 and December 2019 at MH and FSH, were extracted from the prospective databases of the two hospitals, retrospectively reviewed, and inserted in the REMEDY database. Patients with proximal jejuno-ileal or colonic locations were excluded, even if those locations were not suitable for surgery. Variables included in the REMEDY database were age, gender, Montreal classification, perianal disease, smoking habit, family history of IBD, extraintestinal manifestations, age at diagnosis and surgery, disease duration, preoperative blood exams (hemoglobin, white blood cell count, C-reactive protein, total proteins and albumin), preoperative therapy in the last 12 weeks, indication for surgery (stenosing or perforating), surgical access (open or laparoscopic), mesentery characteristics (thickness and presence of enlarged lymph nodes), length of resection, type of anastomosis (manual or stapled), duration of surgery, Clavien–Dindo complications, hospitalization, 90-day readmission, endoscopic and ultrasonographic follow-up evaluation within one year from surgery, post-operative adjuvant treatment, and long-term surgical recurrence [32]. Patients operated on at ASST Fatebenefratelli-Sacco with mesentery removal constitute Group A, while Group B consisted of patients from the Mauriziano hospital whose mesentery was retained. 

## 4. Statistical Analysis

Proportions were compared using two-tailed, Fisher’s exact or chi-square test where appropriate, and continuous variables were analyzed using a two-tailed, unpaired, Student’s t-test. Time-to-event estimates were performed using the Kaplan and Meier function and compared with the Log-rank test. Follow-up was considered complete at time of surgical recurrence, while patients were censored at time of last follow-up visit without recurrence. Cox proportional hazard regression model was used to simultaneously exploring, in relation to time, the effect of independent variables on surgical recurrence. Results were expressed as Hazard Ratios (HR) with 95% Confidence Intervals (CI).

Significance was considered for *p* value ≤ 0.05. Statistical analysis was performed using STATISTICA 8 (data analysis software system, Stat Soft Inc.,Tulsa, OK, USA). 

The ethics committee of MH and FSH approved data auditing and the study has been reported according to the Strengthening the Reporting of Observational Studies in Epidemiology [STROBE] guidelines [33].

## 5. Results

The REMEDY study consisted of 326 patients operated on for complicated ileocecal CD, between January 2009 and December 2019, with a median follow-up of 4.7 ± 3 years. Excision of the mesentery was performed at FSH in 204 patients (62%-Group A) and in 122 patients (38%) ileocolic resection was performed flush with the intestine at MH (Group B). 

In Table 1 are reported the differences between groups in terms of preoperative characteristics, intraoperative findings, and postoperative complications. The two groups were similar in terms of gender, perianal disease, smoking habit, family history, extraintestinal manifestations, age at diagnosis, age at surgery, disease duration, preoperative blood exams (except albumin level), and preoperative therapy, but not combined therapy in the last 12 weeks before surgery (combination of steroids with biologicals or immunosuppressants) and surgical access (open or laparoscopic). The two groups were also similar in the indication for surgery, mesentery thickness, lymph node enlargement, length of resection, and duration of surgery. In Group A, Montreal Classification A2 patients (68.6% vs. 55.7%) where more common than A1 (5.9% vs. 12.3%) and A3 (25.5% vs. 32%, *p* = 0.03), the preoperative level of Albumin was higher (36 ± 5.9 vs. 34 ± 6.8 g/L, *p* = 0.005), combined therapy was less frequent (4.4% vs. 10.7%, *p*= 0.001), and laparoscopic approach more common (93.1% vs. 73.8%, *p* = 0.0001) than in the Group B. Postoperative course in terms of complications, hospital staying and 90-days readmission was also similar in the two groups. 

Postoperative adjuvant therapy was based on immunosuppressants (Azathioprine, 6-Mercaptopurine, or Methotrexate) or biologics (Infliximab, Adalimumab, or Vedolizumab) in 180 patients (55.2%) and on salicylates, antibiotics, budesonide, or no therapy in 146 patients (44.8%). Patients in Group A received immunosuppressants or biologics in 65% (117 patients) while those in Group B in 59.5% (87 patients) without statistically significant difference (*p* = 0.3).

Figure 1 shows endoscopic and ultrasonographic recurrences observed between 6 and 12 months during follow-up. A Rutgeerts’ score between i2 and i4, indicating endoscopic recurrence, was present in 91 patients in Group A (44.6%) and in 57 patients in Group B (46.7%). Ultrasonographic evaluation was available in 290 out of 326 patients (89%) and recurrence, corresponding to a bowel wall thickening ≥4 mm, was observed in 40.4% of patients in Group A and in 41.1% in Group B. A statistical significance was not reached in endoscopic nor ultrasonographic recurrence (*p* = 0.7 and *p* = 0.9, respectively).

Figure 2 reports the 5-year time-to-event estimates for surgical recurrence of all the patients (4%) and the differences, compared using the Log-rank test, between patients with normal or thickened mesentery (3% and 4%, *p* = 0.6), patients submitted to resection or sparing of the mesentery (2.8% and 4%, *p* = 0.6), and patients treated in postoperative adjuvant setting with biological or immunosuppressant versus mesalamine therapy (1.7% vs. 5.4%, *p* = 0.02).

Cox’s proportional hazard regression model results are reported in Table 2. Perforating disease Behavior (B3) resulted as a risk factor related to postoperative surgical recurrence (HR 2, *p* = 0.04), while postoperative adjuvant treatment with biological or immunosuppressants showed a protective rule (HR 2, *p* = 0.04). 

## 6. Discussion

The aim of the REMEDY was to evaluate whether the resection of the mesentery during primary ileocolic resection for complicated CD has an impact on postoperative complications, endoscopic and ultrasonographic recurrences, and long-term surgical recurrences. 

The major drawback of the study is that it is a retrospective analysis, even if performed on prospectively collected data. In particular, since no classification exists in the literature for the characteristics of the mesentery in CD patients, a cut-off of ≥5 mm was used to define mesentery thickening, independent from the presence of wrapping fat, and the presence of three or more lymph nodes in the mesentery of the terminal ileum, greater than 4 mm, for the definition of lymphopathy. This threshold was set on the basis of the large experience of the two centers in IBD surgery and was collected prospectively in both databases.

Furthermore, little retrospective data are available in the literature to compare with on this topic, and even simple aspects as concern the prevalence of mesenteric thickening and enlarged lymphnodes, the correlation among mesentery and patients’ phenotypes, the adequate treatment during surgery, or the correlation with post-operative surgical recurrence are still to be clarified [15]. Another major problem with long-term surgical recurrence is that a consensus on standardized postoperative evaluation is still lacking. In fact, if the Rutgeerts’ score gives information on anastomotic recurrence, on the one hand there is considerable inter-observer variability with this method, and on the other hand the characteristics of the whole bowel wall and of the mesentery are not considered [34].

The two groups were similar for most pre-operative characteristics, but a slight difference was present in the age at diagnosis of the Montreal Classification, with Group B having more A1 and A3 patients (*p* = 0.03). However, in a large, retrospective analysis on 1272 patients from FSH, Sampietro et al. showed an association between mesentery thickening and older age at diagnosis, but not with the age of Montreal Classification [15]. In the present study, no differences were present in the mean age at diagnosis between the Groups. Group B also showed lower pre-operative Albumin levels and a higher number of patients submitted to surgery under steroids or combined medical treatment (any association of steroids with biologicals or immunosuppressants), both well known risk factors for post-operative complications in CD patients, but no differences between groups were present in terms of Clavien–Dindo Classification [15,17,18,20,35]. It has to be noticed that whether the mesentery was resected or retained, the complications were not also affected, suggesting that mesentery resection, considered a major issue by surgeons due to the potential risk of bleeding and complications, is not a technical problem, but a matter of experience and appropriateness [15]. From a technical point of view, the two statistically significant differences were that all the ileo-colic anastomosis in Group B were performed manually compared to only 28.9% in Group A (*p* < 0.0001), and that laparoscopic approach was adopted in 93.1% of patients in Group A compared to 73.8% of Group B (*p* = 0.0001). However, whether manual or stapled, all the anastomoses were performed side-to-side, isoperistaltic at MH and functional end-to-end (anisoperistaltic) at FSH, following the indications of the most recent guidelines [16,18]. In fact, it seems that wide anastomotic diameter is an important discriminating factor for complications and recurrence, whichever anastomotic technique used. In two meta-analyses, one based on eight comparative studies in CD patients, and one based on seven randomized controlled trials in colorectal surgery from the Cochrane Database of Systematic Reviews, functional end-to-end anastomosis were reported with a lower leakage and overall postoperative complications rate than hand-sewn end-to-end (but not side-to-side) anastomosis. Manual or stapled end-to-end, end-to-side, and side-to-side with double blind stumps are associated with a worse long-term recurrence rate and higher postoperative complication rate [36,37,38,39,40]. There is recent evidence that anastomotic configuration probably plays a pivotal role in CD recurrence. In fact, the functional end-to-end handsewn anastomosis proposed by Kono and colleagues (Kono-S) seems to significantly reduce clinical, endoscopic, and surgical recurrence compared even to other side-to-side anastomosis, but it is intriguing to note that in Kono’s technique the mesentery is leaved in situ as it is for strictureplasty [41,42]. 

Endoscopic and ultrasonographic recurrence rates were similar whereas the surgical treatment of the mesentery was different (Figure 1). The two groups were also similar in terms of post-operative adjuvant treatment, since in both hospitals an approach based on ileocolonoscopy (and ultrasonography when available) within one year after surgery, the stratification of risk factors for postoperative recurrence, and early treatment with immunosuppressants or biologics were used [17,43]. 

This strategy, based on the application of current guidelines in postoperative treatment, is likely to be the basis of the long-term results reported in Figure 2. The overall surgical recurrence rate at 5 years was quite low (4%), and it did not seem to be affected by the presence of a thickened or normal mesentery (3% and 4%, *p* = 0.6), nor by its resection or retention (2.8% and 4%, *p* = 0.6). Looking at those patients receiving biological or immunosuppressive therapy, the recurrence rate drops to 1.7%, compared to 5.4% of those undergoing other treatments (*p* = 0.02), as reported in recent European series focused on limited ileocolic disease (Figure 2) [44]. In Cox’s proportional hazard model, mesenteric thickening and lymph node enlargement, as well as mesentery resection, failed to show a role in the surgical recurrence related risks (Table 2). These results are in accordance with those by Sampietro et al. where thickened mesentery and enlarged lymph nodes showed a relation with ileal location (L1) and perforating behavior (B3), and in the Cox’s proportional hazard model, a perforating disease behavior (B3), but not thickened mesentery and enlarged lymph nodes, was associated with an increased risk of surgical recurrence over time [15].

Recently, Coffey et al. reported the first retrospective study suggesting the clinical relevance of including the mesentery in ileocolic resection for CD [9]. The authors compared two cohorts of patients by means of long-term surgical recurrence rate: Group A (30 patients who underwent ileocolic resection without mesentery removal) and Group B (34 patients in which the affected mesentery was fully dissected and partially excised). As a result, cumulative surgical recurrence rate at five years was 40% and 2,9% in Group A and B, respectively. The group from Limerick has the merit of carrying out a thorough study on mesentery anatomy, demonstrating that mesenteric abnormalities strongly relate to mucosal and mural abnormalities of CD, but substantial drawbacks were present in their study, such as the small number of patients included (with the consequent impossibility to perform a multivariate analysis), an historical cohort too heterogeneous in terms of patients’ characteristics and medical treatment, and a proposed mesenteric activity index that is very intriguing, but not yet validated. In particular, the huge difference between the two cohorts could be explained with a different postoperative medical treatment, especially considering the positive effect showed by immunosuppressive and biological adjuvant therapy in the REMEDY [9,11,12]. In addition to the REMEDY results, and in contrast with Coffey’s, another unclear element is represented by the mesentery behavior in the site of a strictureplasty, where the surgeons leave the mesentery untouched. As reported by several authors, the diseased bowel treated by various techniques of strictureplasties has been shown to return to normal within one year after surgery, as well as the contiguous mesentery, with evidence of radiographic, endoscopic, histologic, and cytokine production normalization [20,45,46,47,48]. 

## 7. Conclusions

In conclusion, the REMEDY results support the data from Sampietro et al. that mesenteric thickening and lymph node enlargement are not present in all the patients, including those undergoing surgery for CD of the terminal ileum (prevalence between 66% and 71%), and do not affect post-operative complications [15]. The resection of the mesentery does not seem to reduce endoscopic and ultrasonographic recurrence rate within one year from surgery and long-term surgical recurrence rate at 5 years. Weather the mesentery is resected or retained, early immunosuppressive or biological adjuvant therapy has a significant impact in preventing surgical recurrence. The question as to which is the most important aspect the clinicians have to consider for preventing recurrence, whether the anastomotic configuration, the mesentery resection, the adjuvant therapy, or a score combining these and other risk factors, still remains unanswered and needs further studies. Based on this evidence, the actual wave of “mesentery targeted” surgery needs further studies to be validated and cannot be currently recommended in all patients and for all surgeons.

## Figures and Tables

**Figure 1 jcm-11-01961-f001:**
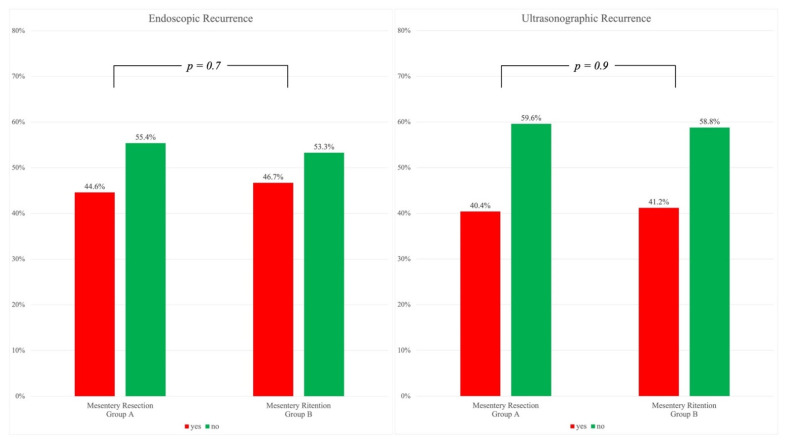
Endoscopic (**left**) and Ultrasonographic (**right**). Recurrences without statistically significant differences at chi-square test.

**Figure 2 jcm-11-01961-f002:**
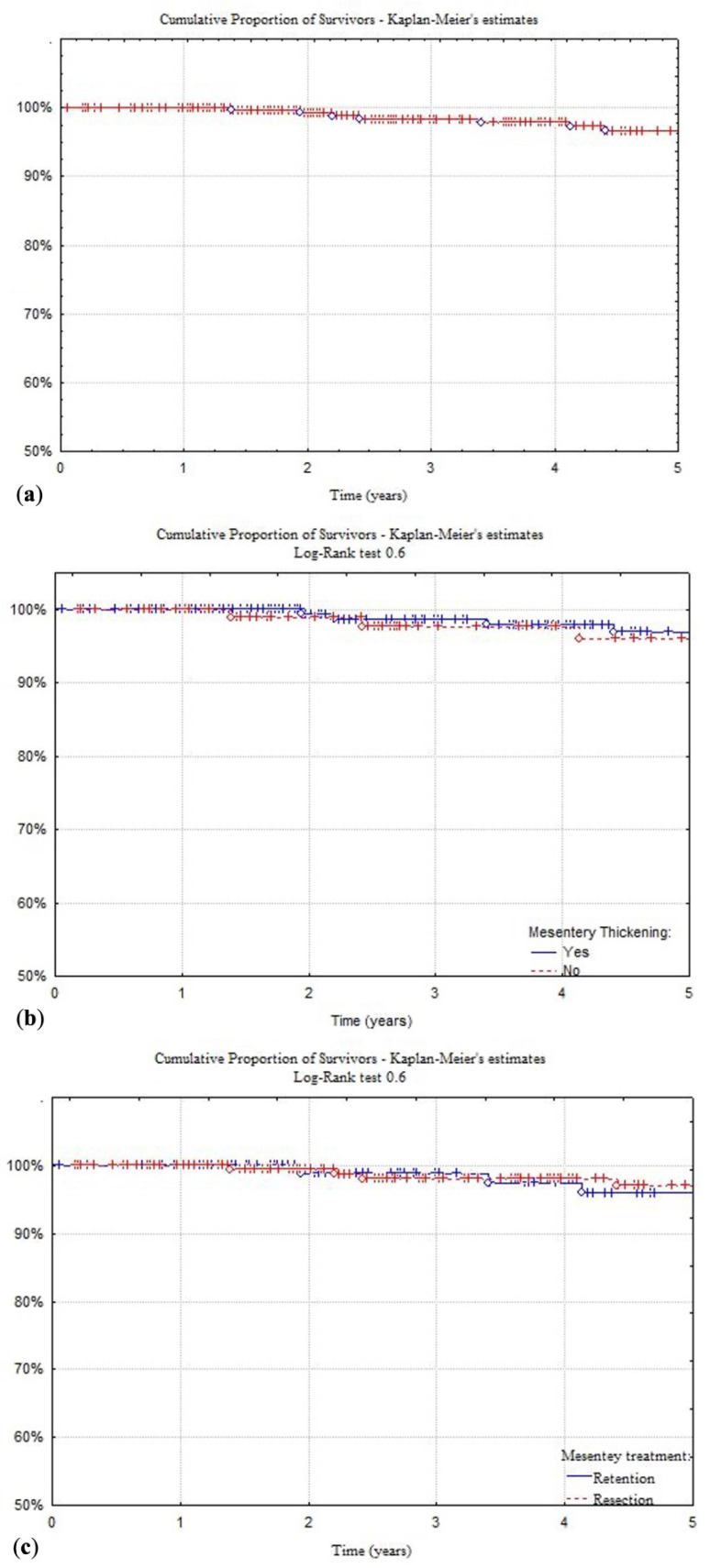
Kaplan and Meier time-to-event estimates of study general population (**a**), patients presenting mesentery thickening (**b**), patients treated with or without mesentery resection (**c**), and with biological or immunomodulator versus mesalamine therapy (**d**). Univariate analysis performed using the Log-Rank test.

**Table 1 jcm-11-01961-t001:** Patients’ history and characteristics. ^¶^ Montreal Classification for CD. ^§^ Azathioprine, 6-Mercaptopurine, or Methotrexate. ^‡^ Any combination of steroids with biologicals or immunosuppressants.

	Group A (*n* = 204) [%] Mesentery Resection	Group B (*n* = 122) [%] Mesentery Preservation	*p*
Gender Male Female	121 [59.3%] 83 [40.7%]	70 [57.4%] 52 [42.6%]	0.8
Age ^¶^ A1 (<16 years) A2 (17–40 years) A3 (>41 years)	12 [5.9%] 140 [68.6%] 52 [25.5%]	15 [12.3%] 68 [55.7%] 39 [32%]	0.03
Behaviour ^¶^ B2 (stricturing) B3 (penetrating)	67 [32.8%] 137 [67.2%]	40 [32.8%] 82 [67.2%]	1
Perianal Disease History ^¶^	27 [13.2%]	16 [13.1%]	1
Smoking Habit	75 [36.8%]	35 [28.7%]	0.1
Family History	14 [6.9%]	4 [3.3%]	0.2
Extraintestinal Manifestations	19 [9.3%]	10 [8.2%]	0.8
Age at Diagnosis (years ± sd)	33.1 ± 13.8	33.4 ± 15.6	0.3
Age at Surgery (years ± sd)	40.5 ± 14.7	40.7 ± 16	0.9
Disease Duration (years ± sd)	7.5 ± 8.3	7.7 ± 8.6	0.8
Preoperative Blood Exams			
Haemoglobin (g/L)	12.6 ± 1.8	12.4 ± 1.9	0.3
WBC Count (×10^9^ L)	7.7 ± 2.7	7.4 ± 2.7	0.3
C-Reactive Protein (g/L)	2.6 ± 4.3	3.3 ± 4.7	0.1
Total Proteins (g/L)	67.2 ± 7	67.6 ± 7.6	0.6
Albumin (g/L)	36 ± 5.9	34 ± 6.8	0.005
Pre-operative Therapy			
Washout/5-ASA	114 [55.9%]	46 [37.7%]	
Steroids	28 [13.7%]	34 [27.9%]	
Immunosuppressants ^§^	21 [10.3%]	13 [10.6%]	
Biologicals	32 [15.7%]	16 [13.1%]	
Combined therapy ^‡^	9 [4.4%]	13 [10.7%]	0.001
Indications for Surgery Stenosis Abscess and/or Fistula	169 [82.8%] 35 [17.2%]	92 [75.4%] 30 [24.6%]	0.1
Surgical Access Open Laparoscopic	14 [6.9%] 190 [93.1%]	32 [22.2%] 90 [73.8%]	0.0001
Mesentery Thickness	135 [66.2%]	87 [71.3%]	0.3
Lymphnodes Enlargement	135 [66.2%]	79 [64.7%]	0.81
Length of Resection (cm ± sd)	24 ± 14	27 ± 16	0.07
Type of anastomosis Manual Stapled	59 [28.9%] 145 [71.1%]	122 [100%] 0 [0%]	<0.0001
Duration of Surgery (minutes ± sd)	150 ± 54	146 ± 55	0.5
Complications (Clavien-Dindo) Grade I II III IV V	11 [5.4%] 20 [9.8%] 16 [7.8%] 1 [0.5%] -	7 [5.7%] 16 [13.1%] 11 [9%] 1 [0.8%] -	0.8
Hospitalization (days ± sd)	8.5 ± 5	9 ± 4	0.3
Readmission 90 days	6 [3%]	6 [4.9%]	0.3

**Table 2 jcm-11-01961-t002:** Multivariate analysis using the Cox’s proportional hazard model. ^¶^ Montreal Classification for CD. ^†^ Indication for fibro-stenotic or perforative disease. ^§^ Patients that completed postoperative induction regimen with a biological therapy or immunosuppressive drug and were able to maintain the treatment for at least 6 months or 1/3 of their follow-up duration, compared to other treatments.

	Hazard Ratio	95% CI	Wald’s Statistics	*p*
Gender	1.5	1.1–1.9	2.3	0.1
Age ^¶^ A1 (vs. A2 and A3)	1.1	0.2–1.8	0.1	0.9
Behaviour ^¶^ B3 (vs. B2)	2	1.1–2.3	4	0.04
Presence of Perianal Disease	1.2	0.7–1.4	1.3	0.2
Active Smoking Habit	0.9	0.5–1.1	0.8	0.3
Lymphnodes Enlargement	0.3	0.2–0.4	1.3	0.6
Mesentery Thickening	0.4	0.2–1.1	1.2	0.7
Mesentery Resection	1.6	1.1–2	2.7	0.09
Indication for Surgery ^†^	1.8	1.2–2.3	3.5	0.06
Manual vs Stapled Anastomosis	1.3	0.8–1.6	1.2	0.2
Postoperative Therapy (protective) ^§^	2	1.8–2.1	4	0.04

## Data Availability

The data supporting the findings of this study are available from the corresponding author upon reasonable request.

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
