# Peer review of "Has the Removing of the Mesentery during Ileo-Colic Resection an Impact on Post-Operative Complications and Recurrence in Crohn’s Disease? Results from the Resection of the Mesentery Study (Remedy)"

_jcm, 2022, doi:10.3390/jcm11071961_

Round 1
Reviewer 1 Report
The authors submitted a retrospective bi institutional study regarding with the aim to study recurrence rate with or without ileal mesenteric resection.
The issue is interesting as literature is poor, and it might guide surgeon in their current surgery.
Despite the retrospective design, this study is well written and methodology is appropriate.
However, there are some issue that authors should discuss to further improve the strength of their study:
1 Ethical committee statement is missing. Is the study registered in a trial database?
2 Surgical approach was not included in the multivariate analysis
3 I suggest author to move the limitation statement of the study at the end of the discussion section.
4 A further discussion regarding the election of 5 mm of cut off might be more appropriate… the sentence “completely arbitrary” should be avoided. I’m sure that author based this cut off on some basis, their experience, knowledge from other disease, radiologist impression, etc..etc..
5 A table with the resume of previous publications regarding this issue may further help the reader
6 conclusions section should state a couple of sentence conclusions based only on the result of your study
7 As it was included in the multivariant analysis, I wonder if preoperative and postoperative treatment might represent a bias for recurrence. Could you better discuss this topic? Even if stated that adjuvant treatment is similar, it should be presented in the table, as well.
8 K-M figure should be presented with the number of censured cases.
Author Response
Reviewer 1.
Comments and Suggestions for Authors
The authors submitted a retrospective bi institutional study regarding with the aim to study recurrence rate with or without ileal mesenteric resection.
The issue is interesting as literature is poor, and it might guide surgeon in their current surgery.
Despite the retrospective design, this study is well written and methodology is appropriate.
However, there are some issue that authors should discuss to further improve the strength of their study:
1 Ethical committee statement is missing. Is the study registered in a trial database?
The study is a retrospective analysis on prospective data bases, so the study was not registered as a trial. The approval of database auditing by the two Institutions is reported in the Statistical Analysis paragraph.
2 Surgical approach was not included in the multivariate analysis
Multivariate analysis is Cox’s regression model that explores, in relation to time, the effect of independent variables on surgical recurrence. Surgical approach, in particular open vs laparoscopic technique, is not a time-dependent variable associated to long-term recurrence. Anastomosis configuration, on the contrary, has been reported in the literature having a role on post-operative recurrence and thus was analysed in the Cox’s model
3 I suggest author to move the limitation statement of the study at the end of the discussion section.
Mesentery treatment is an “hot topic” of surgery for IBD and the REMEDY has some relevant limitations. We feel it would be relevant for readers having these limitations in mind during their reading of the discussion.
4 A further discussion regarding the election of 5 mm of cut off might be more appropriate… the sentence “completely arbitrary” should be avoided. I’m sure that author based this cut off on some basis, their experience, knowledge from other disease, radiologist impression, etc..etc..
We agree with the reviewer that the cut off of the mesentery thickness was based on the experience of thousands of surgical procedures, but the definition of “arbitrary” was only a methodological definition. A change was made in the text according to reviewer’s suggestion.
5 A table with the resume of previous publications regarding this issue may further help the reader
The number of publications on this topic is very limited. The group from Limerik has published some redundant papers, but all the few relevant articles were cited in the text where consistent with the discussion.
6 conclusions section should state a couple of sentence conclusions based only on the result of your study
This suggestion is unclear to the authors, since the most relevant results of the REMEDY are cited in the conclusions paragraph
7 As it was included in the multivariant analysis, I wonder if preoperative and postoperative treatment might represent a bias for recurrence. Could you better discuss this topic? Even if stated that adjuvant treatment is similar, it should be presented in the table, as well.
Preoperative therapy, considered as those medical treatments the patient has taken in the last 12 weeks before surgery, was considered for the possible impact on postoperative complications and thus inserted in the appropriate Table (Table 1). Preoperative treatment has never proved to have any relation with post-operative recurrence since surgery is supposed to make a sort of “refresh” of medical history. However, the reviewer’s comment turned on a light in Author’s mind. It would be interesting to analyse the medical history of CD patients before surgery to check if somehow it has an relation with long-term surgical recurrence. Unfortunately, the data bases used in the study are surgical data bases and start their observation when a surgical indication is given during MDM. We are available for a collaboration on this topic.
Postoperative adjuvant therapy is reported in detail in the Results chapter, including statistical difference.
8 K-M figure should be presented with the number of censured cases.
The program used for statistical analysis do not have this kind of output. So, if this is not mandatory for the reviewer, we would like to leave the graphic output unchanged.
Reviewer 2 Report
This is an interesting manuscript detailing an the importance of mesenteric resection in Crohn Disease postoperative complications. Nowadays, this is a recurrent topic in IBD surgery with high interest.
The study is well written and with an adequate number of patients.
My main concern with this manuscript is regarding the evaluation of the mesentery retrospectively (as also the authors assess) and a completely arbitrary cut-off of ≥5mm for mesentery thickening. However, it is necessary to establish a method to evaluate retrospectively the IBD patients mesentery and this seems to be an accurate one.
Author Response
Reviewer 2.
Comments and Suggestions for Authors
This is an interesting manuscript detailing on the importance of mesenteric resection in Crohn Disease postoperative complications. Nowadays, this is a recurrent topic in IBD surgery with high interest.
The study is well written and with an adequate number of patients.
My main concern with this manuscript is regarding the evaluation of the mesentery retrospectively (as also the authors assess) and a completely arbitrary cut-off of ≥5mm for mesentery thickening. However, it is necessary to establish a method to evaluate retrospectively the IBD patient’s mesentery and this seems to be an accurate one.
A comment on the decision to set the cut off value of the mesentery at 5mm has been made in the text, also according to Reviewer 1’s indication. However, since this is a retrospective analysis performed in two prospective data bases, this value war decided at the beginning of data auditing and thus it was collected prospectively ever since.